# An Overview of the Potential for Pharmacokinetic Interactions Between Drugs and Cannabis Products in Humans

**DOI:** 10.3390/pharmaceutics17030319

**Published:** 2025-03-01

**Authors:** Dolly Andrea Caicedo, Clara Pérez-Mañá, Magí Farré, Esther Papaseit

**Affiliations:** 1Department of Clinical Pharmacology, Hospital Universitari Germans Trias I Pujol, Institut de Recerca Germans Trias i Pujol (IGTP), Carretera de Canyet, s/n, 08916 Badalona, Spain; dacaicedo.mn.ics@gencat.cat (D.A.C.); cperezm.mn.ics@gencat.cat (C.P.-M.); epapaseit.germanstrias@gencat.cat (E.P.); 2Department of Pharmacology, Therapeutics and Toxicology, Universitat Autònoma de Barcelona (UAB), Bellaterra, 08193 Cerdanyola del Vallès, Spain

**Keywords:** cannabis, tetrahydrocannabinol, THC, cannabidiol, CBD, drug interactions, medicines, pharmacokinetics

## Abstract

Cannabis is the most commonly used illicit substance worldwide. Recent years have seen an increase in cannabis consumption, and with new approvals and therapeutic indications, there are challenges in minimizing the risks and interactions between cannabis-based products, cannabis prescription drugs, other approved prescription drugs, and other substances of abuse. Thus, identifying the enzymes metabolizing cannabinoid drugs and their relationship with other prescription drugs is crucial for understanding the potential interactions and effects of their simultaneous use. This article offers a comprehensive review of cannabis and the pharmacokinetic interactions between cannabis products, cannabis prescription drugs, and other approved prescription drugs, as well as other substances of abuse. It also compiles existing evidence of these interactions and describes the clinical outcomes associated with the inhibition or induction of various enzymes.

## 1. Introduction

During the last years, an increased number of countries (states) have legalized the medical use of cannabis and/or its recreational use. Because the legalization and social acceptance of cannabis continue to expand, it is crucial to understand how these substances can impact public health and clinical practice. One relevant issue is the possibility of interaction with other medicines and its consequences.

### 1.1. Cannabis

*Cannabis sativa* L., an herbaceous plant belonging to the Cannabaceae family, is globally recognized as “marijuana” and “hemp”. It consists of a single species, “sativa”, which encompasses several subspecies or varieties, such as *Cannabis sativa* ssp. *sativa*, *Cannabis sativa* ssp. *indica*, *Cannabis sativa* ssp. *ruderalis*, and *Cannabis sativa* ssp. *afghanica* [1].

The cannabis plant harbors over 100 cannabinoids and a plethora of other compounds, such as terpenoids, flavonoids, and omega-3 and 6 fatty acids. Delta-9-tetrahydrocannabinol (THC) and cannabidiol (CBD) are the most important cannabinoids, widely studied and sought after by consumers, along with their acidic forms. A crucial biochemical process in cannabis involves the conversion of cannabigerolic acid (CBGA) into tetrahydrocannabinolic acid (THCA) and cannabidiolic acid (CBDA), which are inactive acidic forms of the cannabinoids THC and CBD, naturally occurring in the cannabis plant. Under high temperatures (as in smoking or vaporizing) or through decarboxylation, THCA can convert into ∆-9-THC or its isomer ∆-8-THC. Additionally, THCA can oxidize to form cannabinolic acid (CBNA), which converts to cannabinol (CBN). On the other hand, CBDA can convert into CBD [2].

THC, the main psychoactive compound in cannabis, is recognized for its reinforcing and addictive properties, which result from its interaction with the endocannabinoid system. This system is primarily composed of cannabinoid type 1 receptors (CB1R) and cannabinoid type 2 receptors (CB2R). CB1R is mainly found in the central nervous system (CNS), and CB2R predominantly in the immune system, hematopoietic cells, and certain brain regions [3,4]. THC primarily targets CB1R, as a partial agonist, eliciting the characteristic “high” associated with marijuana use. In contrast, CBD acts chiefly as an antagonist of CB1R and as a non-competitive negative allosteric modulator of CB2R, and also interacts with various other receptors, thereby not eliciting the characteristic effects of THC, including its abuse potential [5,6]. Consequently, CBD has shown analgesic, anti-inflammatory, and neuromodulatory properties, suggesting its therapeutic potential [7,8,9].

### 1.2. Cannabis Products

Cannabis products are used for both “medical” and “non-medical” purposes, and the latter include recreational and industrial applications. To differentiate them, various terms are employed: “medical cannabis”, “recreational cannabis”, and “industrial hemp” [10].

While most cannabis use is recreational, the concept of legal medical cannabis use has gained increasing attention over the past few decades and has been legislated in various countries [11,12]. Since the mid-1990s, several states in the United States of America (USA) have legalized medical cannabis for various symptoms and illnesses, including multiple sclerosis, chronic pain, and terminal cancer. In 1999, Canada introduced a medical cannabis program. Since the early 2000s, many countries have progressively legislated the medical use of cannabis under specified conditions [13]. Over the last two decades, there has been a resurgence in the use of cannabis and cannabinoids to treat a variety of health conditions, such as chemotherapy-induced nausea/vomiting, chronic pain, and spasticity associated with multiple sclerosis and refractory epilepsy, with potential efficacy in treating post-traumatic stress disorder (PTSD) [14].

Currently, the term “medical use of cannabis” refers to the practice of permitting cannabis use for medical purposes or the utilization of cannabis-based products (Table 1), cannabis-derived compounds, or cannabis-related compounds (Table 2). “Medical cannabis products” encompass a wide array of preparations and pharmaceutical forms, offering various routes of administration, each containing different active cannabinoid compounds with distinct regulatory and medical implications [15]. Presently, nabilone (Cesamet^®^, a synthetic cannabinoid), CBD (Epidiolex^®^, a natural extract), nabiximols (Sativex^®^, a natural blend of THC/CBD), and dronabinol (Marinol^®^, Syndros^®^, synthetic THC) are the only pharmaceutical cannabinoid drugs approved for prescription use [12,16]. These are authorized for treating secondary muscle spasticity in multiple sclerosis, refractory epilepsy associated with Dravet syndrome and Lennox–Gastaut syndrome, antiemetic therapy in oncological patients, chronic pain, glaucoma, and anorexia with weight loss in patients with human immunodeficiency virus (HIV) [17,18,19].

### 1.3. Epidemiology

Cannabis remains the most widely used illicit recreational drug worldwide [20]. The cannabis industry has expanded significantly due to increasing legal access. In 2022, the global population of drug users reached 292 million, making a 20% rise over the past decade. Cannabis continues to be the most commonly used drug worldwide, with 228 million individuals reported as users [21].

The potency of cannabis products in the market has increased in recent years. From 2011 to 2021, the average potency of herbal cannabis in the European Union increased by 57%, and the potency of cannabis resin increased by nearly 200% [22]. The FDA reports that the rapidly expanding cannabinoid market raises safety concerns, particularly due to the increased consumption of formulations with high concentrations of emerging semi-synthetic cannabinoids, such as delta-8-tetrahydrocannabinol (Δ-8-THC), delta-10-tetrahydrocannabinol (Δ-10-THC), or hexahydrocannabinol (HHC). This trend may pose potential risks to public health, including increased co-use with other cannabis derivatives, other substances of abuse like nicotine, and concurrent medications [23].

Epidemiological data on medical cannabis usage are limited, with prevalence estimates varying widely, based on the context and population of interest [13,22]. Most prevalence studies do not specify the type of cannabis preparation or administration route. A cross-sectional study in Washington State (USA), examining the prevalence and clinical characteristics of documented medical cannabis use in electronic primary care health records (n = 185,565), revealed that approximately 2% of patients had documented medical cannabis use and 20% had other cannabis use. Those using medical cannabis had a higher prevalence of conditions for which cannabis use might confer potential benefits (chronic pain, multiple sclerosis, muscle spasticity, severe nausea, sleep disorders) at 49.8%, compared to 39.9% for patients with other types of use. Notably, chronic pain was the most common condition among medical cannabis users at 35.4% [24]. Another prevalence study with 27,169 participants from Canada and the USA found a medical cannabis usage rate of 27%, highest among young adults. States with legal recreational cannabis use showed a prevalence of 34% versus 23% in states where it was illegal. The most common therapeutic purposes were chronic pain (53%), sleep difficulties (46%), headaches/migraines (35%), appetite disturbances (22%), and nausea/vomiting (21%). For mental health, the most common uses were for anxiety (52%), depression (40%), and PTSD (17%). Additionally, 11% used cannabis to manage alcohol or other substance consumption [25]. A retrospective study across 33 clinics in the United States, with 61,379 patients, found that 44.2% used prescribed cannabis. The most common medical conditions treated were chronic pain (38.8%), anxiety (13.5%), and PTSD (8.4%). Other conditions included back and neck problems (6.5%), arthritis (3.9%), insomnia (3.4%), and cancer-related pain (2.7%). Migraines, depression, attention-deficit, muscle spasms, fibromyalgia, chronic nausea, epilepsy, and headaches have been reported as the primary conditions in 2% or fewer cases [26]. There is a high prevalence of CBD use among individuals with physical and mental health issues, justifying the need for public health warnings about potential adverse effects and drug interactions [27].

In reference to the scope of this review, two studies have been published, based on case reports and data from the FDA Adverse Event Reporting System Database (FAERS), that explore the interactions between cannabis-derived products and medications in pediatric and adult populations. In individuals under 18 years of age, an increased risk of severe adverse reactions was observed when cannabinoids were combined with medications such as methadone, everolimus, fluoxetine, and paroxetine [28]. In adults, the medications most involved in potential interactions were anticonvulsants, antidepressants, warfarin, and tacrolimus. A higher proportion of severe events, including deaths, were found when cannabis was used in combination with controlled substances (medications with strict regulation, such as opioids, benzodiazepines, and certain stimulants) compared to non-controlled substances [29].

### 1.4. Pharmacokinetics

Understanding the pharmacokinetic profile of cannabis is crucial to determine its potential interactions with other approved prescription drugs. The pharmacokinetics of cannabis vary depending on the formulation, the method, and route of administration [30,31,32].

#### 1.4.1. Absorption

The primary routes of cannabis administration are smoking and vaporization. THC and CBD can be detected in plasma within seconds following initial inhalation, with peak concentrations (Cmax) observed between 1.2 and 30 min. The bioavailability of THC post-inhalation varies from 10% to 35%, whereas the bioavailability of CBD ranges from 11% to 45%. The quantity absorbed is contingent upon the number of inhalations, duration of each puff, and depth of inhalation [30,31,33].

Inhalation and oromucosal administration of cannabis circumvent or diminish first-pass metabolism, which is typically seen with the oral administration of cannabis [33]. Oromucosal preparations, such as buccal aerosols like nabiximols, are quickly absorbed, leading to higher plasma concentrations of THC than those achieved orally, although lower than those from inhalation. Such preparations are beneficial for symptoms that require swift alleviation [31,34].

THC (dronabinol) and CBD are lipophilic and have low oral bioavailability [22]. Oral bioavailability is estimated to be approximately 5–20% due to erratic absorption. They undergo gastric destruction and first-pass hepatic metabolism, reaching a maximum plasma concentration (Tmax) at 1–5 h after oral administration. Effects begin within 30 min of ingestion and can last for up to 6 h post-ingestion [32,34]. Given this profile, oral formulations are beneficial for patients who require extended symptom relief [31].

Transdermal administration of cannabis circumvents first-pass metabolism. Cannabis has a highly hydrophobic nature that impedes diffusion through the aqueous layer of the skin. Studies have shown that CBD has greater skin permeability than THC [31,32,35]. Research has documented that, following the application of a dermal patch (Δ-8-THC 16 mg/mL in propylene glycol–water–ethanol, in a 1:1:1 ratio), the mean steady-state plasma concentration of Δ-8-THC reached approximately 4.4 ng/mL at 1.4 h and was sustained for at least 48 h. Additionally, the permeabilities of CBD and cannabinol (CBN) were 10 times higher than those of Δ-8-THC [36]. Although transdermal administration is not currently employed clinically, it has potential for future use [35].

#### 1.4.2. Distribution

Cannabinoids generally exhibit high liposolubility, enabling them to easily cross the blood–brain, placental, and breast tissue barriers. They bind to plasma proteins at approximately 95%, primarily to lipoproteins and albumin, and accumulate in the liver, lungs, and fatty tissues, leading to multicompartmental pharmacokinetics [30,31,32]. The volume of distribution (Vd) of THC and CBD in adults varies from 2.5 to 10 L/kg, assuming a body weight of 70 kg [30,37].

#### 1.4.3. Metabolism

The two primary cannabinoids, CBD and THC, undergo extensive metabolism in the liver, primarily by cytochrome P450 isozymes CYP2C9, CYP2C19, and CYP3A4. In humans, these processes involve hydroxylation or oxidation, followed by glucuronidation catalyzed by UDP-glucuronosyltransferase (UGT) enzymes [30,37,38].

THC (dronabinol) undergoes first-pass metabolism in the liver, transforming into 11-hydroxy-THC (11-OH-THC), primarily through CYP2C9. This metabolite is active and slightly more potent than ∆-9-THC. Additionally, 11-OH-THC crosses the blood–brain barrier more readily. After a second hepatic metabolism, 11-OH-THC is converted into various inactive metabolites (CYP2C9), including 11-nor-carboxy-∆9-THC (THC-COOH), which subsequently binds to glucuronic acid to produce THC-COO-Glucuronide; these inactive metabolites are primarily excreted through urine. Other metabolic pathways involve CYP3A4 and UGT enzymes, with more than 80 distinct THC metabolites being identified [5,30,37,38]. Figure 1 shows the metabolism of THC in a simplified way.

CBD is primarily metabolized by the isoenzymes CYP2C19 and CYP3A4, with additional metabolism by CYP1A1, CYP1A2, CYP2C9, and CYP2D6. The UGT isoforms responsible for the phase 2 conjugation of CBD are UGT1A7, UGT1A9, and UGT2B7. Following hydroxylation by CYP2C9 to form 7-hydroxy CBD (7-OH-CBD), other hepatic metabolism occurs to form 7-carboxy-cannabidiol (7-COOH-CBD), the primary inactive metabolite of CBD, as well as other metabolites such as 6-hydroxycannabidiol (6-OH-CBD). These metabolites are then excreted primarily via feces, with a lesser amount excreted in urine [5,31,39]. Figure 2 shows the metabolism of CBC in a simplified way.

#### 1.4.4. Elimination

THC is primarily excreted through feces (approximately 65–80%) and urine (about 20–35%). Following THC consumption, urine predominantly contains acidic metabolites, such as THC-COOH, mainly in its glucuronidated form, whereas feces contain both neutral and acidic metabolites, with THC-COOH (28%) and 11-OH-THC (20%) being the most prevalent. In contrast, the majority of CBD is excreted unchanged in feces (33%), with both unchanged CBD and CBD glucuronide excreted in the urine [28,31,33,40].

The elimination half-life of THC is highly variable, and is characterized by a rapid initial phase and a relatively short intermediate half-life (approximately 4 h), followed by a significantly prolonged terminal elimination phase, with half-lives ranging from 24 to 36 h or more. This extended duration is attributed to the slow redistribution of THC from deep compartments, such as fatty tissues. As a result, THC concentrations exceeding 1 μg/L can be detected in the blood of heavy users for more than 24 h after the last cannabis use. Similarly, CBD has been reported to exhibit a long terminal elimination half-life, with values of 24 ± 6 h after intravenous dosing, 31 ± 4 h post-inhalation, and 2–5 days following repeated daily oral administration [31,41]. For occasional cannabis users, traces of metabolites can be found in urine for up to a week after consumption, whereas for those who use it daily, these metabolites may remain detectable for as long as a month [32].

### 1.5. Pharmacogenetics/Pharmacogenomics

Polymorphisms in genes encoding receptors, enzymes, and transporters can influence the pharmacokinetics, efficacy, and resistance to cannabinoid therapy.

Multiple genetic polymorphisms have been identified in phase 1 metabolic enzymes (CYP2C9, CYP3A4, and CYP2C19) and phase 2 metabolic enzymes (UGT1A3, UGT1A9, UGT1A10, and UGT2B7) that affect the metabolism of cannabinoids. These polymorphisms can result in either rapid (ultrarapid metabolizers) or slow (poor metabolizers) enzymatic metabolisms. Interactions between cannabinoids and other drugs are complex. For instance, individuals with CYP2C9*2 or CYP2C9*3 variants, especially those homozygous for CYP2C9*2 or heterozygous/homozygous for CYP2C9*3, may experience significantly reduced THC and CBD metabolism. Poor metabolizer phenotypes indicate a decreased rate of THC conversion into its active metabolite 11-OH-THC, resulting in a higher concentration ratio of THC/11-OH-THC. However, since both 11-OH-THC and THC possess psychoactive properties, altering their ratio is expected to have a minimal impact on the onset of psychotic symptoms, but the decreased formation of inactive metabolites (THC-COOH) enhances the risks associated with cannabis use [42]. In a retrospective study of 71 patients, it was suggested that atypical genetic variants (in CYP2C9, CYP2C19, CYP3A4, and CYP3A5) and concomitant medications may alter the metabolism of THC and CBD. However, the lack of clinical data makes it difficult to estimate their impact [43].

Considering the genetic variations, certain individuals or populations may be at a heightened risk of toxicity and adverse events. The label information for dronabinol indicates that individuals with genetic variants associated with diminished CYP2C9 function could experience a two-to-three-fold increase in dronabinol exposure. Moreover, individuals with genetic glucuronidation disorders, such as Gilbert’s syndrome, may exhibit increased serum bilirubin levels and should be treated cautiously when administered nabiximols according to label [39,44,45,46].

### 1.6. Acute, Chronic Effects and Intoxication

The impact of cannabis preparations varies with their THC and CBD content. This variation underpins the differing effects of medicinal cannabis products, which typically contain higher levels of CBD and lower levels of THC than recreational cannabis [16].

THC primarily induces psychoactive effects, such as pleasure, relaxation, and euphoria, along with cognitive alterations, including impaired memory and attention, diminished psychomotor and cognitive performance, and altered perception of time passage [16,17,18,19]. Sensory perception is often enhanced; however, feelings of increased well-being may lead to anxiety, dysphoria, or panic [16,17,19,47] Common physiological effects of cannabinoids include tiredness, tachycardia, dizziness, dry mouth, orthostatic hypotension, conjunctival injection, decreased lacrimation, increased appetite, and muscle relaxation [48]. Acute impairments in psychomotor performance, such as anxiety and attention deficits, have been associated with driving impairment, heightening the risk of traffic accidents and hindering the operation of heavy machinery, particularly when combined with alcohol [4,17,18].

Regular cannabis inhalation can lead to chronic respiratory symptoms, including a cough and chronic obstructive pulmonary disease. It has also been associated with an elevated risk of cardiovascular disorders, such as acute myocardial infarction, stroke, and transient ischemic attack. Furthermore, chronic consumption may increase the likelihood of mental health disorders, including depression, schizophrenia, and other psychoses. Cannabis use is a critical factor in acute psychosis and is associated with a high incidence of schizophrenia. The risk of mental illness escalates with increased frequency and duration of use, and early initiation of cannabis consumption is a significant risk factor [18,47]. Other nonspecific cannabis-related disorders include flashbacks, memory impairment, and amotivational syndrome [17,49]. THC can modify the secretion of sex hormones [18]. Cannabinoid hyperemesis syndrome (CHS) is associated with chronic cannabis use [50].

Long-term cannabis consumption can lead to substance use disorder (addiction). Tolerance to the undesired effects of cannabinoids, such as dizziness, tiredness, cardiovascular, and psychoactive effects, may develop over days or weeks. Withdrawal symptoms occur only in heavy cannabis users after the abrupt cessation of use [44,51]. Abstinence can present with symptoms including irritability, nervousness, anxiety, insomnia, vivid dreams, decreased appetite, weight loss, depressive disorder, craving for cannabis, headache, restlessness, tremors, chills, and sweating [51]. However, withdrawal symptoms are rarely problematic in the controlled medical administration of cannabinoids [47,52].

According to the DSM-5 manual, a diagnosis of cannabis intoxication requires a recent history of cannabis use and clinically significant behavioral or psychological changes following consumption, such as euphoria, impaired judgment, and compromised motor skills. Additionally, at least two of the following symptoms should be present within approximately two hours after cannabis use: red eyes, dry mouth, increased appetite, and elevated heart rate. These symptoms must not be attributable to any other medical or psychiatric condition [4,49].

THC is primarily sought by recreational users in social settings and is often referred to as a “gateway drug”. This designation suggests that individuals who use it may be at an increased risk for other substance use disorders, such as those involving cocaine or opioids [17,49].

Medications with similar effects to those of cannabis can potentiate these effects. For instance, sedatives such as benzodiazepines, when used concomitantly with cannabis, may augment sedative effects. Likewise, cannabis use can exacerbate the adverse effects of medications with similar side effects, such as additive tachycardia, when cannabinoids and atropine are taken together. In some cases, additive effects are beneficial, and medicinal cannabis may be administered concurrently with antispasmodics, bronchodilators, antiemetics, and analgesics [16,47].

## 2. Data Search and Selection Methods

We utilized the PubMed database, using the search strategy: (Cannabis OR medical cannabis OR THC OR tetrahydrocannabinol OR CBD OR cannabidiol OR synthetic cannabinoids OR dronabinol OR Marinol OR Syndros OR nabilone OR Cesamet OR Canemes OR Epidyolex OR Bedrocan OR nabiximols OR Bedrobinol OR Bediol OR Bedica OR Bedrolite OR Sativex) AND (interaction OR drug interaction). We only selected human data written in English. The filters applied included Clinical Trial, Humans, and English. The search, performed on 12 January 2024, yielded 275 articles.

Additionally, data from the interactions section of the Summary of Product Characteristics/Prescriber Information for cannabis products were reviewed and included in the Stockley’s Drug Interactions book. For interaction risk rating, we relied on the UpToDate search platform (UpToDate, https://www.uptodate.com, accessed on 2 September 2024). Furthermore, we classified the quality of evidence and clinical relevance based on previous systematic review papers on pharmacokinetic interactions [53,54],

We classified the evidence as follows:Strong: the interaction is supported by evidence from at least one meta-analysis, systematic review, or a clinical trial (randomized or non-randomized);Moderate: the interaction is supported by evidence from at least one observational study (cohort or case-control) or a minimum of three case reports;Weak: the interaction is supported by evidence from fewer than three case reports.

And we classified clinical relevance as follows:Major: the interaction poses a risk of harm or injury. The parameter variation is 400% or more [Internationalized Normalized Ratio (INR), transaminases, Area Under the Curve (AUC)] or at least 80% (clearance);Significant: the interaction requires closer monitoring. The parameter variation is between 100 and 400% (INR, transaminases, AUC) or between 50% and 80% (clearance);Minor: the interaction causes little to no harm. The parameter variation ranges between 25% and 100% (INR, transaminases, AUC) or between 20% and 50% (clearance);Lack of relevance and/or not applicable (not classified in previous definitions).

## 3. Cannabis Products’ Drug Interactions

Pharmacokinetic interactions occur when a cannabis product is combined with another substance, such as a medicine or drug of abuse, leading to altered concentrations that may result in changes in the effects. Conversely, pharmacodynamic interactions arise when the effects of pharmacological targets are altered. In both scenarios, it is crucial to consider the implications of efficacy and safety.

Cannabinoids can act as either precipitant drugs, which induce changes in the concentrations or effects of another drug, or as an object drug, whose concentrations or effects are altered by other substance. Cannabinoids, when mainly acting as precipitant drugs, can enhance the bioavailability or decrease the clearance of other medications via enzymatic inhibition. This may lead to heightened systemic exposure, and consequently, an elevated risk of adverse effects.

Research on the risks of interactions between drugs and cannabis products has mainly focused on THC and CBD, and, therefore, the potential risk of other cannabinoids is still unknown.

As previously noted, THC and CBD act as substrates, primarily for CYP2C19, CYP2C9, and CYP3A4, and as modulators, functioning as inhibitors or inducers of various CYP450 isoenzymes. These isoenzymes are crucial for the metabolism of various medications. While primary pharmacokinetic interactions can be attributed to inhibition, induction, or shared CYP metabolism between cannabinoids and other drugs, interactions involving UGT enzymes or P-gp have also been reported. In vitro studies have revealed that cannabinoids and the major metabolites of THC strongly inhibit several P450 enzymes, including CYP2B6, CYP2D6, CYP1A2, and CYP2C9. CBD exhibits inhibitory competition with CYP3A4, CYP2C9, CYP2D6, CYP2B6, and CYP2E1, whereas CBN also inhibits CYP2C9, CYP2B6, and CYP2E1 [5,11]. Additionally, CBD is a weak inhibitor of CYP1A2 (label) and P-glycoprotein (P-gp) [55].

When considering combined in vitro studies examining the induction and inhibition of the major isoforms of human CYP450 by THC, CBD, and CBN, a low risk of clinically significant interactions with the majority of medications is suggested [48]. However, those involving drugs with a narrow therapeutic index (NTI) as object drugs or strong inhibitors/inducers as precipitant drugs are more likely to be clinically significant. Additionally, cannabinoids are highly protein-bound, which may lead to displacement interactions with other protein-bound drugs, potentially increasing the free fraction of NTI medications and altering their pharmacokinetics. Kocis and Vrana describe a list of at least 60 medications with a NTI [53,56]. Table 3 shows how cannabinoid medications affect the metabolism of substrate medication (object) with NTI, based on the medication label. In Table 4 we present medications with NTI and cannabinoid medications with high protein binding.

Nabiximols (Sativex^®^, THC: CBD; 1:1, equal quantities) can produce metabolic interactions, mainly mediated by its CBD component. Nabiximols was observed to be a reversible inhibitor of CYP3A4, CYP2C9, CYP2C19, CYP1A2, and CYP2B6 at concentrations significantly higher than those likely to be clinically achieved. In vitro research also demonstrated the potential of nabiximols for the time-dependent inhibition of CYP3A4 at clinically relevant concentrations. The inactivation rate of CYP3A4 is expected to be rapid. In an in vitro study, nabiximols inhibited the UGT enzymes, UGT1A9 and UGT2B7, at concentrations achievable in clinical settings [39]. Plasma concentrations of CBD and THC from clinical doses of nabiximols may be sufficient to induce CYP3A4, CYP2B6, and CYP1A2 at the mRNA level, potentially inducing drug-metabolizing enzymes and transporters in vitro; however, the clinical relevance of this induction remains unknown [56].

Dronabinol, a synthetic form of THC (see THC sections), has an inhibitory potential on enzymes that is not fully understood; however, it is suspected that there may be potential drug–drug interactions (DDI) with CYP2C9 and CYP3A4 substrates. CYP3A4 inhibitors may increase the systemic exposure of dronabinol and/or its active metabolite [56]. The liquid formulation of dronabinol (Syndros^®^) contains 50% dehydrated alcohol, which can cause a disulfiram-like reaction with medications such as disulfiram or metronidazole, leading to symptoms like abdominal cramps, nausea, and flushing [45].

Evidence indicates that nabilone (Cesamet^®^, synthetic cannabinoid) is extensively metabolized by multiple P450 enzyme isoforms. Nabilone exhibits a weak inhibitory effect on CYP3A4 and CYP2E1, and a moderate inhibitory effect on CYP2C9 and CYP2C8 [44,56].

CBD (Epidiolex^®^) may induce or inhibit the metabolism of other medications that are substrates of CYP1A2 and CYP2B6, resulting in either a decrease or an increase in the effect of the other medication. Additionally, CBD can inhibit the metabolism of other medications by the enzymes CYP2C9, CYP2C19, CYP2C8, and by the UGT2B7 and UGT1A9 enzymes. Figure 3 summarizes the influence of CBD, dronabinol, nabiximols, and nabilone on the activity of different enzymes.

CBD is purported to mitigate the adverse effects of THC; however, studies in humans and dogs present conflicting evidence suggesting that CBD may inhibit the metabolism of Δ9-THC and 11-OH- THC, increasing the adverse events of THC. These may be influenced by the ratio of the compounds, administered dose, and individual responses [51,57,58]. In a randomized, crossover, double-blind clinical trial, 18 healthy adults underwent experimental sessions involving the oral administration of brownies with varying cannabinoid contents: a placebo with no cannabis extract, a Δ9-THC-dominant extract (20 mg Δ9-THC without CBD), and a CBD-dominant extract (20 mg Δ9-THC + 640 mg CBD). The CBD-dominant extract led to a higher Cmax and a larger area under the plasma concentration-time curve for Δ9-THC-COOH, Δ9-THC, and 11-OH-Δ9-THC compared to the Δ9-THC-dominant extract at an equivalent Δ9-THC dose, resulting in more pronounced adverse effects, such as anxiety, sedation, and memory difficulties [57]. Conversely, other studies have indicated that CBD minimally affects THC concentrations in saliva when both compounds are vaporized at a 1:1 ratio [59]. Studies using functional magnetic resonance imaging in adolescent and young adult cannabis users have found no evidence that CBD attenuates the effects of THC [60]

Given the escalating use of cannabis products and their enzymatic metabolism, it is pertinent to examine the potential pharmacokinetic interactions between these products and medications [5,11,61]. Considering that cannabis is often used concurrently with other drugs of abuse, this review incorporates a section on interactions with such substances.

### 3.1. Interactions with Medicines

Here we describe some of the main interactions classified by pharmacological groups. For further information about interactions classified by pharmacological groups, please refer to Table 5, which summarizes the interactions found between cannabis and other medications, with a description of the observed pharmacokinetic effects, clinical effects, the risk rating (based on the UpToDate classification: A, B, C, D, or X) and if it is documented in the prescribing information, evidence (strong, moderate, weak), and clinical relevance (major, significant, minor, lack of relevance, and/or not applicable).

#### 3.1.1. Anticoagulants

In general, there is no evidence to support an interaction between cannabinoids and anticoagulants, except for warfarin. Warfarin is primarily metabolized by CYP2C9 and CYP3A4. CBD inhibits the CYP2C9 and CYP3A4 enzymes, and, therefore, co-administration can lead to the accumulation of warfarin, increasing INR levels and the risk of bleeding. Close monitoring of INR is recommended during the initiation and adjustment of CBD dosage. Regarding THC, there is an in vitro study that shows THC inhibits the CYP2C9-mediated metabolism of warfarin [62,63,64].

#### 3.1.2. Antiepileptics

Antiepileptic drugs represent one of the primary classes of medications involved in pharmacological interactions with cannabis. Generally, the combination of antiepileptic drugs and CBD leads to an increase in the plasma levels of these medications, especially in the case of eslicarbazepine due to the inhibition of CYP2C19. Although no severe adverse effects have been reported, monitoring plasma levels is recommended when coadministered with CBD.

Stiripentol through CYP2C19 inhibition and valproate via an unknown mechanism (possible UGT1A9 and UGT2B7 inhibition) may interact with CBD. CBD may increase stiripentol levels by 28% for Cmax and 55% for AUC, resulting in variable adverse effects, including rashes and elevated liver enzymes (AST/ALT). Elevated liver enzyme levels have also been observed with the coadministration of CBD and valproic acid, even with normal valproic acid levels, possibly due to a hepatotoxic interaction [39,65]. Therefore, active monitoring of side effects, including liver function tests, is recommended, considering a dose reduction of valproate/stiripentol and CBD [66,67,68]

For some CNS depressants used as anticonvulsants and barbiturates, it has been documented that THC decreased clearance of these agents, presumably through competitive inhibition of metabolism [44,45,46].

There is significant evidence of interaction between CBD and clobazam (see benzodiazepine section).

#### 3.1.3. Analgesics

THC has been shown to reduce the clearance of antipyrine (phenazone), likely through the competitive inhibition of metabolism [44,45,46,69]. Globally, the use of this analgesic has significantly decreased due to the increased prevalence of acetaminophen and ibuprofen as preferred alternatives used in some countries.

#### 3.1.4. Opioids

Buprenorphine is primarily metabolized by the enzyme CYP3A4. Recreational cannabis use has been found to inhibit CYP3A4, leading to increased concentrations of buprenorphine, thereby increasing the risk of intoxication [70].

The use of cannabis with oxycodone or morphine may enhance analgesic effects, but no significant changes in the area under the curve of concentrations have been revealed to explain this occurrence. It has been suggested that the improved analgesia might be due to cannabis-induced slowing of gastrointestinal motility, resulting in a sustained release effect of morphine. However, there is no conclusive evidence of this interaction [71]. No evidence of interaction between CBD and fentanyl IV has been found [72]. In patients with cannabis use disorder undergoing opioid substitution therapy, a higher risk of drug interactions has been identified, predominantly with levomethadone, buprenorphine, and morphine. Therefore, clinical and therapeutic monitoring is recommended to enhance treatment safety [73].

#### 3.1.5. Benzodiazepines

There is evidence of a significant interaction between clobazam and CBD. CBD inhibits the enzyme CYP2C19 involved in the metabolism of the active metabolite of clobazam, n-desmethylclobazam [66,74,75,76,77,78]. It has been found that levels of N-desmethylclobazam can increase three- to four-fold when combined with clobazam. Additionally, clobazam may increase exposure to 7-OH-CBD, with plasma AUC potentially increasing by 47% [39,74]. This interaction leads to an increase in n-desmethylclobazam levels, the prolongation of elimination half-life, and an elevated risk of adverse effects such as sedation and somnolence [53].

#### 3.1.6. Immunosuppressants and Proliferation Inhibitors

Interactions have been identified between cannabinoids and mTOR inhibitors (sirolimus and everolimus), as well as with calcineurin inhibitors (tacrolimus), involving the inhibition of CYP3A4 and P-glycoprotein/ABCB1. Significant increases in everolimus and sirolimus levels have been found following CBD treatment [79]. CBD may increase everolimus exposure of approximately 2.5-fold for both Cmax and AUC [39,80].

CBD use has been associated with elevated tacrolimus levels, leading to adverse events including nausea, dry mouth, dizziness, heat episodes, and severe toxicity with encephalopathy [81,82,83,84]. However, there are inconsistent outcomes in some patients, which may be explained by interindividual variability [53,83]. A phase I study evaluated the interaction between CBD and tacrolimus in 12 healthy participants. The study found that CBD increased tacrolimus Cmax 4.2-fold and AUC_0_-∞ 3.1-fold (*p* < 0.0001) without affecting its half-life. These findings suggest that CBD inhibits CYP3A4-mediated metabolism of tacrolimus, which may require dose reduction and frequent therapeutic monitoring in transplant patients using CBD [85].

#### 3.1.7. Selective Serotonin Reuptake Inhibitors (SSRIs)

Citalopram and escitalopram are metabolized through CYP2C19 and CYP3A4, both of which are inhibited by CBD. Some cases have demonstrated increased plasma concentrations of citalopram with concurrent use of CBD [86]. Data are limited, and further studies are needed on this topic [67].

#### 3.1.8. Tricyclic Antidepressants

Cases of tricyclic antidepressant toxicity after smoking cannabis have been described; however, serum levels were not measured [53,87,88].

#### 3.1.9. Anti-Infectives

Ketoconazole acts as a strong CYP3A4 inhibitor, resulting in increased concentrations of THC (increase in Cmax and AUC of THC, 1.2- and 1.8-fold, respectively) and CBD (increase in Cmax and AUC of CBD, 2- and 2-fold, respectively). Therefore, reducing the dose of ketoconazole is recommended when administered with CBD. Conversely, rifampicin acts as a CYP3A4 inducer. When used concurrently with cannabis, decreases in concentrations of both CBD (reductions in Cmax by 50% and AUC by 60%) and THC (reductions in Cmax by 40% and AUC by 20%) have been observed, necessitating a gradual increase in CBD dose for patients experiencing minimal effects [39,65,67].

Fluconazole acts as a moderate inhibitor of CYP3A4 and a strong inhibitor of CYP2C9, and may increase the systemic exposure of dronabinol and/or its active metabolite, increasing the mean THC Cmax by 22% and mean AUC by 32%. Exposure to the metabolite 11-OH-THC also increased approximately 2.1-fold for Cmax and 2.5-fold for AUC. The Cmax of CBD increased by approximately 40% with fluconazole. All of this can increase cannabis-product-related adverse reactions [39,44,45,46].

#### 3.1.10. Antiretrovirals

Antiretroviral drugs such as indinavir and nelfinavir have shown potential interaction (possible CYP3A4 and CYP2C induction). A 17.4% decrease in the Cmax of indinavir and 14,1% of nelfinavir has been observed in the presence of smoked cannabis [67,89,90].

#### 3.1.11. Proton Pump Inhibitors (PPIs)

Although omeprazole inhibits CYP2C19, no significant alteration in plasma levels of THC and CBD has been found [65].

#### 3.1.12. Psychostimulants

It has been suggested that CBD inhibits enzymes other than CYP, such as CES1 (serine hydrolase), which is involved in the metabolism of drugs like methylphenidate (MPH). However, the co-administration of CBD and MPH has been found to result in insignificant pharmacokinetic changes. Nonetheless, further studies are suggested to evaluate long-term effects [67,91].

#### 3.1.13. Antipsychotics

The administration of smoked cannabis has been documented to increase the clearance of chlorpromazine by 50%, while the co-administration of tobacco and cannabis further increases this clearance by 107% (tobacco smoke acts as an enzyme inducer). Regarding clozapine, a 50% increase in plasma levels has been observed 2–4 weeks after cessation of cannabis consumption (tobacco smoke produces CYP1A2 induction, and clozapine is a substrate of CYP1A2). This can increase sedation and dizziness after stopping smoking, and, therefore, dose adjustments of clozapine are recommended after stopping smoking cannabis [92,93].

#### 3.1.14. Methylxanthine Derivatives

Theophylline and other methylxanthine derivatives are known for common DDI. CBD has been found to inhibit the enzyme CYP1A2 (weak inhibitor), which is responsible for the metabolism of caffeine (theophylline and caffeine are CYP1A2 substrates). This leads to a substantial increase in caffeine’s AUC and half-life, consequently heightening the risk of adverse effects [94]. CBD may increase caffeine exposure by 15% for Cmax and 95% for AUC compared to when caffeine is given alone [39,94].

### 3.2. Interactions with Drugs of Abuse

Table 6 presents the key interactions between cannabinoid products and drugs of abuse. It details the precipitant substance, observed alterations in cannabinoid pharmacokinetics, and primary clinical effects. Furthermore, the table displays cannabinoids and the observed changes in the pharmacokinetics of the substance of abuse, along with its main clinical effects.

### 3.3. Pharmacodynamic Interactions (Medicines and Drugs of Abuse)

Cannabis products exhibit additive CNS effects when combined with CNS depressants such as anticonvulsants, barbiturates, ethanol, benzodiazepines, lithium, buspirone, opioids, muscle relaxants, and antihistamines [39,44,45,46]. The symptoms of this interaction may include dizziness, confusion, sedation, and somnolence. Additionally, impairments in attention, judgment, thinking, and psychomotor skills may be exacerbated.

Additionally, additive cardiac effects such as hypertension, hypotension, tachycardia, and syncope have been observed when combined with amphetamines, cocaine, and other sympathomimetics, as well as hypotensors or anticholinergic agents. The concurrent use of amoxapine, amitriptyline, desipramine, or other tricyclic antidepressants can exacerbate drowsiness, hypertension, and tachycardia due to the combined beta-adrenergic and antimuscarinic effects of tricyclics with the beta-adrenergic effects of cannabis. Additive tachycardia and drowsiness may also occur with atropine, scopolamine, some antihistamines, and other anticholinergic agents [34,39]. Moreover, smoking cannabis with disulfiram or fluoxetine can also induce hypomanic reactions. Dronabinol inhibits serotonin uptake in a manner similar to selective serotonin reuptake inhibitors (SSRIs) [92].

Combining cannabis with a nicotine patch has additive effects on the heart rate. Moreover, the effects of oral THC are enhanced by opioid receptor blockade with naltrexone. Cannabis counteracts the stimulant and hyperthermic effects of ecstasy (3,4-methylenedioxymethamphetamine, midomafetamine) [92]. Smoked cannabis plus cisplatin increases the risk of stroke, but the mechanism is unknown [92].
pharmaceutics-17-00319-t005_Table 5Table 5Interactions found between cannabis and other medications, including quality of evidence, clinical relevance, risk rating based on the UpToDate classification (A: no interaction, B: no action needed, C: monitor therapy, D: modify regimen, or X: avoid combination) and pharmacokinetic interactions in the prescribing information (Yes/No). Abbreviation: PI, prescription information. NA, not applicable. For other abbreviations, refer to the text (ref., references).DrugCannabinoid and Pharmacokinetics MechanismObserved Pharmacokinetic Findings and EffectsClinical EffectsRecommendationRisk Rating/PIEvidence**Clinical Relevance****Ref.**AnticoagulantsWarfarinCBD, Nabilone, Dronabiol: inhibits the CYP2C9 and CYP3A4 enzymes/Highly protein bound -No clinical trial with cannabis has been conducted. Out of a total of 10 reported cases, 9 describe variations in INR (90%). A maximum INR variation of +0.4 to +9.61 was described in 8 of the 10 reports.May increase the risk of bleeding.Adjust warfarin dosage and closely monitor INR.C/NoModerateMajor[62,63,95]AntiepilepticsEslicarbazepineCBD: UnknownEslicarbazepine as an object and CBD as a precipitantAdverse effects were reported without clear evidence linking them to eslicarbazepineAdjust eslicarbazepine dosage and, if it is possible, monitor plasma levelsC/NoStrongMinor[66,96]-In adult subjects, trough concentrations of eslicarbazepine were elevated. The mean baseline concentration was 14.4 ± 7.4 μg/mL, increasing with CBD in subsequent measurements to 16.8 ± 7.9 μg/mL and 17.8 ± 9.1 μg/mL, demonstrating statistically significant changes.-Co-administration of eslicarbazepine and CBD increased eslicarbazepine level by 23% in one patient, while no changes were observed in the other patient.StiripentolCBD: CYP2C19 inhibitionStiripentol as an object and CBD as a precipitantAdditive CNS depressant effectsAdjust stiripentol dosage and if it is possible monitor plasma levelsC/YesStrongMinor[39,65,74]-CBD slight increase in stiripentol levels, with C_max_ and the area under the concentration-time curve over the dosing interval (AUCtau) elevated by 2500 ng/mL (32.1%) and 27,000 ng · h/mL (57.6%), respectively.-CBD may increase stiripentol levels by 28% for C_max_ and 55% for AUC, potentially leading to diverse adverse effects such as rashes and elevated liver enzymes (AST/ALT).CBD as an object and stiripentol as a precipitant medication-Minor reductions in CBD major metabolites, 7-OH-CBD which a reduction of 29% and 7-COOH-CBD which decreased by 13%. However, CBD was not affected.ValproateCBD: Unknown/Possible UGT1A9 and UGT2B7 inhibition/Highly protein bound-AST/ALT levels were significantly higher in participants co-administered with valproate and CBD, with mean AST and ALT levels of 37.1 U/L and 35.3 U/L, respectively. In contrast, participants not taking valproate had lower AST and ALT levels of 23.97 U/L and 23.7 U/L.May increase ALT and AST levelsAssess liver function before starting CBD and monitor liver functionC/NoStrongMinor[66]Analgesics—OpioidsBuprenorphineCannabis recreational: CYP3A4 inhibitionBuprenorphine as an object and CBD as a precipitantMay increase the risk of sedation and somnolenceAdjust buprenorphine dosage and monitor plasma levels of buprenorphineC/NoModerateSignificant[70]-Buprenorphine concentrations were found to be 170% higher in individuals who concurrently use cannabis recreationally. In one case report, a patient experienced a 95% decrease in serum buprenorphine levels upon discontinuing cannabis use.MethadoneCBD: CYP3A4 and CYP2C19 inhibitionMethadone as an object and CBD as a precipitantMay increase fatigue and somnolenceAdjust methadone dosage and if it is possible monitor plasma levelsC/NoWeakSignificant[97]-CBD can inhibit methadone metabolism. The serum methadone levels were measured at 271 ng/mL (2 days after discontinuing CBD), 149 ng/mL (7 days after discontinuing CBD), and 125 ng/mL (14 days after discontinuing CBD)FentanylCBD: unknownCBD as an object and fentanyl as a precipitant medicationProbably not clinically significantAdjust fentanyl dosage and, if it is possible, monitor plasma levelsC/NoStrongLack of relevance[72]-Plasma concentrations of CBD were not significantly affected.BenzodiazepinesClobazamCBD: CYP2C19 inhibitionClobazam as an object and CBD as a precipitantMay increase the incidence of sedation and somnolenceAdjust clobazam dosage and it is possible, monitor plasma levelsC/YesStrongMinor[39,53,66,74,75,76,77,78] -CBD increase mean concentrations of N-desmethylclobazam (10–526%).-In the general group, there was a small increase in exposure to steady-state clobazam (C_max_ = 20%, AUCτ = 21%) and a notable increase in exposure to N-desmethylclobazam (C_max_ = 3.4-fold [239%], AUCτ = 3.4-fold [238%]). In the epilepsy volunteer group, there were no effects on exposure to clobazam, but there was an increase in exposure to N-desmethylclobazam (C_max_ = 2.2-fold [122%], AUCτ = 2.6-fold [164%])-Slight increase in Clobazam exposure, with a trough C_max_ of 1.20 ng/mL, and an increase in N-desmethylclobazam exposure, where the mean C_max_ increased 3.39-fold. -Elevated levels (3- to 4-fold) of N-desmethylclobazam (substrate of CYP2C19) can occur when combined with CBD.CBD as an object and clobazam as a precipitant medication-Clobazam may increase exposure to 7-OH-CBD, for which plasma AUC increased by 47%. Clobazam showed a slight increase in CBD exposure, with a trough C_max_ of 1.34 ng/mLBrivaracetamCBD: CYP2C19 inhibitionBrivaracetam as an object and CBD as a precipitantMild effects, such as diarrhea and somnolence, may occurAdjust brivaracetam dosage and, if it is possible, monitor plasma levelsC/NoStrongSignificant[96]-Brivaracetam levels increased by 107% to 280% in patients receiving co-medication of brivaracetam and CBD.Immunosuppressants and Proliferation InhibitorsTacrolimusCBD: CYP3A4 inhibition/Highly protein boundTacrolimus as an object and CBD as a precipitantNausea, dizziness, and somnolence may appear. Creatinine levels may also increaseAdjust tacrolimus dosage and monitor blood levels of tacrolimus. Dose reduction is warranted when creatinine levels increaseC/NoStrongSignificant[83,84,85]-A case report showed baseline tacrolimus levels ranged from 3.9 to 8.4 ng/mL. By Day 164, the dose-adjusted tacrolimus level had increased approximately threefold to 13.3 ng/mL. Serum creatinine levels rose to 2.4 mg/dL by Day 124 (baseline 1.2 mg/dL), prompting discontinuation of tacrolimus for a week, after which creatinine levels decreased to 1.5 mg/dL. Attempts to maximize tacrolimus dosage led to a subsequent increase in creatinine levels by Day 282.-Increased tacrolimus’ Cmax 4.2-fold and AUC_0_-∞ 3.1-foldEverolimusCBD: CYP3A4 inhibitionEverolimus as an object and CBD as a precipitantMay increase the incidence of diarrheaAdjust everolimus dosage and monitor blood levelsC/NoModerateMinor[79]-Median increase of everolimus AUC by 9.8 ng/mL as compared with baselineSirolimus (convencional)CBD: CYP3A4inhibitionSirolimus as an object and CBD as a precipitantMay increase the incidence of diarrheaAdjust sirolimus dosage and, if it is possible, monitor blood levelsD/NoModerateMinor[79]-Median increase of sirolimus AUC by 5.1 ng/mL as compared with baselineCyclosporineCBD: CYP3A4 inhibition/Highly protein bound drugsInconclusive findings:-CBD might increase and displace the free fraction of other concomitantly administered protein-bound drugs (not confirmed in vivo).-The levels of cyclosporine are stable in co-administration with CBDProbably not clinically significantAdjust cyclosporine dosage and if it is possible monitor blood levelsC/YesStrongLack of relevance[44,45,46,83]Selective serotonin reuptake inhibitors (SSRIs)CitalopramCBD: CYP2C19 and CYP3A4 inhibitionCitalopram as an object and CBD as a precipitantThe reported adverse events were mildAdjust citalopram or escitalopram dosage and, if it is possible, monitor plasma levelsD/NoModerateMinor[86]-Citalopram plasma concentrations increased from baseline (42 ng/mL) to Week 8 (79 ng/mL).Tricyclic antidepressantsImipramineSmoked a marijuana cigarette: unknown-Serum levels were not measured Disorientation, restlessness, dizziness, and palpitations may appear, suggestive of tricyclic antidepressants toxicityAdjust imipramine dosage and, if it is possible, monitor plasma levelsC/NoStrongNA[53,88]AntipsychoticsClozapineCannabis and cigarettes: CYP1A2 inductionClozapine as an object and cannabis and cigarettes as a precipitantMay increase the risk of hallucinations.Adjust clozapine dosage and if it is possible, monitor plasma levelsC/NoWeakSignificant[92,98]-In one case report, a patient stopped the consumption of cannabis and cigarettes, and the plasma levels of clozapine increased by 230%. ChlorpromazineCannabis and cigarettes: CYP1A2 inductionChlorpromazine as an object and cannabis and cigarettes as a precipitantMay increase the risk of sedation and somnolenceAdjust chlorpromazine dosage and if it is possible, monitor plasma levelsC/NoStrongMinor[93]-Increase the clearance of chlorpromazine by 50%, while co-administration of tobacco and cannabis further increases this clearance by 107%.PimozideCBD: CYP3A4 inhibitorsNo clinical trials with cannabis have been conducted. Considering mechanisms, metabolism, and theoretical principles, a significant interaction may be possible.May produce potentiallyserious clinical outcomes, increasing the risk of CNS depressionAvoid combinationX/NoNANA[99]BromperidolNabilone: UnknownNo clinical trials with cannabis have been conducted. Considering mechanisms, metabolism, and theoretical principles, a significant interaction may be possible.May produce potentiallyserious clinical outcomes, increasing the risk of CNS depressionAvoid combinationX/NoNANA[99]MethotrimeprazineDronabinol: UnknownNo clinical trials with cannabis have been conducted. Considering mechanisms, metabolism, and theoretical principles, a significant interaction may be possible.May increase the risk of CNS depressionAvoid combinationX/NoNANA[99]Anti-infectivesKetoconazoleNabiximols: CYP3A4 inhibitionNabiximols as an object and ketoconazole as a precipitantSomnolence, malaise, dizziness, anxiety, and disorientation may appearAdjust ketoconazole dosage and, if it is possible, monitor plasma levelsC/NoStrongMinor[65]-In subjects using THC/CBD oromucosal spray alongside ketoconazole, there were increases in C_max_ levels of THC from 2.65 to 3.36 ng/mL, CBD from 0.66 to 1.25 ng/mL, and 11-OH-THC from 3.59 to 10.92 ng/mL. Overall, C_max_ levels increased by 36% to 87%.RifampicinNabiximols: CYP3A4 induction, CYP2C19 inducersNabiximols as an object and rifampicin as a precipitantFew clinical manifestations; headache can occurAdjust rifampicin dosage and, if it is possible, monitor plasma levelsC/NoStrongMinor[65]-When using THC/CBD oromucosal spray alone, C_max_ levels were observed as 2.94 ng/mL for THC, 1.03 ng/mL for CBD, and 3.38 ng/mL for 11-hydroxy-THC (11-OH-THC). With co-administration of THC/CBD spray and rifampicin, C_max_ levels decreased to 1.88 ng/mL for THC, 0.50 ng/mL for CBD, and 0.45 ng/mL for 11-OH-THC. Overall, C_max_ levels were reduced by 26% to 87%.Fusidic acidDronabinol, Nabilone: CYP3A4 substratesNo clinical trials with cannabis have been conducted. Considering mechanisms, metabolism, and theoretical principles, a significant interaction may be possible.May increase the risk of toxicity of these medicationsModify the regimen of fusidic acid to include aggressive monitoring, empirical adjustments, and consideration of alternative agentsD/NoNANA[99]FluconazoleDronabinol: CYP2C9 and CYP3A4 inhibitorDronabinol as an object and fluconazole as a precipitantCan increase cannabis-product-related adverse reactionsAdjust fluconazole dosage and, if it is possible, monitor plasma levels.C/YesWeakMinor[39,45,46]-May increase the systemic exposure of dronabinol and/or its active metabolite, increasing THC C_max_ by 22% and AUC by 32%. Exposure to the metabolite 11-OH-THC also increased approximately 2.1-fold for C_max_ and 2.5-fold for AUC. -The C_max_ of CBD increased by approximately 40% with fluconazole.MetronidazoleDronabinol (Syndros^®^): UnknownDronabinol as an object and metronidazole as a precipitantDisulfiram-like reaction. May produce potentiallyserious clinical outcomesAvoid combinationX/YesNANA[45]-The liquid formulation (Syndros^®^) of dronabinol contains alcohol and Metronidazole can inhibit the metabolism of alcohol.ItraconazoleNabilone, dronabinol: UnknownNo clinical trials with cannabis have been conducted. Considering mechanisms, metabolism, and theoretical principles, a significant interaction may be possible.Can increase cannabis-product-related adverse reactions Adjust itraconazole dosage and, if it is possible, monitor plasma levelsC/NoNANA[99]FexinidazoleDronabinol, Nabilone: CYP3A4 substratesNo clinical trials with cannabis have been conducted. Considering mechanisms, metabolism, and theoretical principles, a significant interaction may be possible.May produce potentiallyserious clinical outcomesAvoid combinationX/NoNANA[99]Ordinazole, SecnidazoleDronabinol: UnknownNo clinical trials with cannabis have been conducted. Considering the metabolism and theoretical principles, a significant interaction may be possible.May increase the risk of toxicity of these medicationsAvoid combinationX/NoNANA[99]AntiretroviralsIndinavirTHC cigarettes: Possible CYP3A4 inductionIndinavir as an object and cannabis and THC cigarettes as a precipitantProbably not clinically significantAdjust indinavir dosage and, if it is possible, monitor plasma levelsC/NoStrongLack of relevance[89]-Decrease in C_max_ of indinavir by 17.4% (n = 11)NelfinavirTHC cigarettes: Possible CYP3A4 inductionNelfinavir as an object and cannabis and THC cigarettes as a precipitantProbably not clinically significantAdjust nelfinavir dosage and, if it is possible, monitor plasma levelsC/NoStrongLack of relevance[89]-Decrease in C_max_ of nelfinavir by 14.1% (n = 14)Mehtylxanthine derivatesCaffeineCBD: CYP1A2 inhibitionCaffeine as an object and CBD as a precipitantDiarrhea and elevated liver enzymes (AST, ALT, GGT) may appearNo action neededB/YesStrongMinor[39,94]-CBD and caffeine co-administration may increase caffeine exposure by +15% for C_max_ and +95% for AUC compared to a single administration of caffeine.Proton pump inhibitor (PPIs)OmeprazoleTHC and CBD: CYP3A4 inhibitionTHC and CBD as an object and omeprazole as a precipitantDizziness may appearNo action neededB/NoStrongLack of relevance[61,65]-No significant alteration in plasma levels of THC and CBD.PsychostimulantsMethylphenidate (MPH)CBD: possible serine hydrolase inhibitionMPH as an object and CBD as a precipitantProbably not clinically significantNo action neededB/NoStrongLack of relevance[65,91,100]-The geometric mean ratios (GMR) for AUC from time 0 to infinity (AUC_inf_) and C_max_ with CBD co-administration, compared to MPH monotherapy were 1.08 (0.85, 1.37) and 1.09 (0.89, 1.32), respectively. T_max_ was longer for CBD (4 h) than MPH (1.25 h).Other central nervous system depressantsHexobarbitalCBD: possible CYP3A4 inhibitionHexobarbital as an object and CBD as a precipitantMay increase the risk of sedation and somnolenceAdjust hexobarbital dosage and, if it is possible, monitor plasma levelsNot found/NoStrongMinor[101]-CBD reduced hexobarbital clearance by 35%, compared to when it was not administrated, in subjects who consume recreational cannabis regularly.PhenobarbitalDronabinol: possible CYP3A4, CYP2C9 and CYP2C19 inductionPhenobarbital as an object and CBD as a precipitantLoss of efficacy of cannabis productAdjust phenobarbital dosage and, if it is possible, monitor plasma levelsC/YesStrongLack of relevance[39,58,66]-In co-administration with CBD, no significant changes were found in plasma concentrations of phenobarbital.Dronabinol as an object and phenobarbital as a precipitant-May reduce the systemic exposure of dronabinol parent drug and/or its active metabolitePropofolCBD: UGT1A9 and UGT2B7Inconclusive findings:-Concentrations of propofol may increase by enzyme inhibition.-Some studies reported that an increased dose is required to induce anesthesia in cannabis users.May increase the risk of sedation and somnolenceAdjust propofol dosage and, if it is possible, monitor plasma levelsC/YesModerateNA[39,102]Other unclassified drugsDisulfiram Dronabinol (Syndros^®^): UnknownDronabinol as an object and Disulfiram as a precipitantDisulfiram-like reaction. May produce potentiallyserious clinical outcomesAvoid combinationX/YesNANA[45]-The liquid formulation (Syndros^®^) of dronabinol contains alcohol, and disulfiram can inhibit the metabolism of alcohol.Colchicine, Pralsetinib, Relugolix, Estradiol, Norethindrone, Rimegepant, Tizanidine, Ubrogepant, Venetoclax, Digoxin, Lefamulin, Lemborexant, Afatinib, BerotralstatCBD: P-glycoprotein/ABCB1 inhibitorNo clinical trials with cannabis have been conducted, but several pharmacokinetic studies have found that the AUC and C_max_ of these drugs increase during co-administration with P-glycoprotein inhibitors.May increase the risk of toxicity of these medicationsModify the regimen of these drugs to include aggressive monitoring, empirical adjustments, and consideration of alternative agentsD/NoNANA[99,103]Bilastine, Doxorubicin (conventional)Pazopanib, Repotrectinib, Topotecan, Vincristine (liposomal)CBD: P-glycoprotein/ABCB1 inhibitorsNo clinical trials with cannabis have been conducted. but several pharmacokinetic studies have found that the AUC and C_max_ of these drugs increase during co-administration with P-glycoprotein inhibitors.May produce potentiallyserious clinical outcomesAvoid combinationX/NoNANA[99]Amifostine, ObinutuzumabNabilone: UnknownNo clinical trials with cannabis have been conducted. Considering mechanisms, metabolism, and theoretical principles, a significant interaction may be possibleMay increase the risk of toxicity of these medicationsModify the regimen of these drugs to include aggressive monitoring, empirical adjustments, and consideration of alternative agentsD/NoNANA[99]LomitapideCBD: CYP3A4 inhibitorsNo clinical trials with cannabis have been conducted. Considering mechanisms, metabolism, and theoretical principles, a significant interaction may be possible.May increase the risk of toxicity of these medicationsModify the regimen of lomitapide to include aggressive monitoring, empirical adjustments, and consideration of alternative agentsD/NoNANA[99]CilostazolCBD: CYP2C19 inhibitorsNo clinical trials with cannabis have been conducted. Considering mechanisms, metabolism, and theoretical principles, a significant interaction may be possible.May increase the risk of toxicity of these medicationsModify the regimen of cilostazol to include aggressive monitoring, empirical adjustments, and consideration of alternative agentsD/NoNANA[99]MavacamtenCBD: CYP2C19 inhibitorsNo clinical trials with cannabis have been conducted. Considering mechanisms, metabolism, and theoretical principles, a significant interaction may be possible.May produce potentiallyserious clinical outcomesAvoid combinationX/NoNANA[99]FezolinetantCBD: CYP1A2 inhibitorsNo clinical trials with cannabis have been conducted. Considering mechanisms, metabolism, and theoretical principles, a significant interaction may be possible.May produce potentiallyserious clinical outcomesAvoid combinationX/NoNANA[99]
pharmaceutics-17-00319-t005_Table 6Table 6Overview of cannabis drug interactions with drugs of abuse. Abbreviations (see text): HR heart rate, MAP: mean arterial pressure, BP: blood pressure, T: temperature, EEG: electroencephalogram.Drug Cannabinoid and Pharmacokinetics MechanismObserved Pharmacokinetic Findings and EffectsClinical EffectsReferencesAlcoholTHC, CBN and dronabinol: UnknownTHC as an object and alcohol as a precipitantMay increase HRMay increase subjective “like” and its duration, and euphoriaMay increase subjective “drunkenness” May impair driving lateral control performanceMay impair horizontal gaze nystagmus and one-leg studyMay produce additional decremental effects on performance[104,105,106,107,108,109,110,111]-May increase C_max_ of THC, 11-OH-THC and CBNCocaineSmoked cannabis: UnknownCocaine as an object and smoked cannabis as a precipitantMay increase MAP and HRMay increase subjective “high”May reduce reaction time and proficiency of impulse controlMay increase errorsMay decrease subjective “hunger” and “calm”May reduce the latency to cocaine effects and decrease the duration of dysphoric or bad effects[112,113,114,115,116]-May reduce C_max_ of cocaine and benzoylecgonine (BZ) and AUC of BZ-May increase C_max_ and AUC of cocaineDextroamphetamineTHC: UnknownNo available informationMay increase BP and HRMay increase tremors, intensity of symptoms, and their duration. May increase subjective “high”May reduce flexibility to closure[117,118]3,4-metilendioximetanfetamina (MDMA)THC: UnknownNo pharmacokinetics modificationsMay increase BP, HR, and TMay increase subjective drug effects and drug strengthMay impair task performance (EEG oscillations)[119,120,121]Methylphenidate (MPH)CBD: possible serine hydrolase inhibitionNo pharmacokinetics modifications (for more information, see the Psychostimulants section of Table 5).May increase HR and BP May increase subjective “feel drug”, “good effect”, and “take drug again”[65,100]NicotineTHC: UnknownNo pharmacokinetics modifications May increase HRMay increased subjective “euphoria” and “high” and its duration[122,123,124]

## 4. Conclusions

Cannabinoids can interact with therapeutic medications, alcohol, and other substances of abuse. The extensive variety available in the market, combined with the widespread use of cannabinoids for medical conditions, may lead to numerous pharmacokinetic interactions. These interactions predominantly occur within two enzyme families, cytochrome P450 and UGT, as most market medications are metabolized by these enzymes. Despite limited and conflicting data on these interactions, ongoing research is essential. In clinical practice, standardized strategies for managing cannabinoid interactions are lacking; however, practitioners are advised to exercise caution and monitor potential clinical effects that may indicate an interaction with drugs or substances of abuse.

## Figures and Tables

**Figure 1 pharmaceutics-17-00319-f001:**
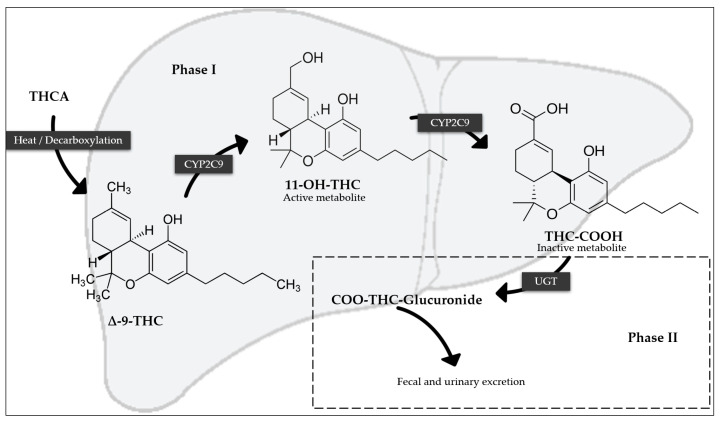
Main metabolism of THC (regarding abbreviations, see text).

**Figure 2 pharmaceutics-17-00319-f002:**
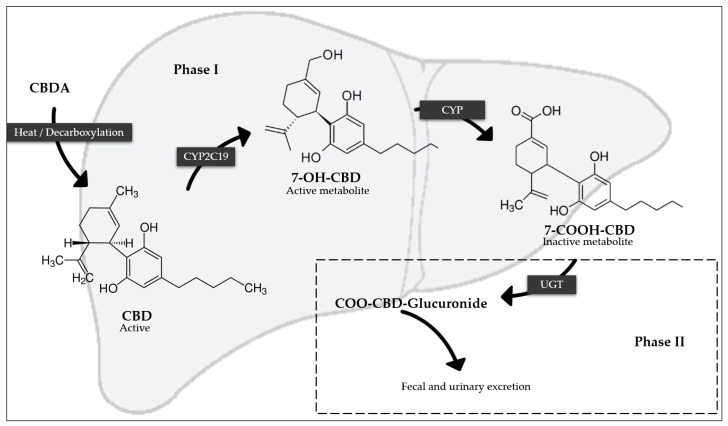
Main metabolism of CBD (regarding abbreviations, see text).

**Figure 3 pharmaceutics-17-00319-f003:**
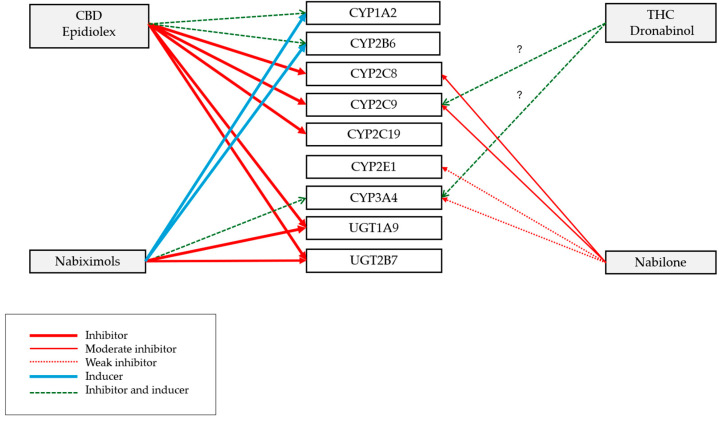
Influence of cannabinoids on enzyme activity based on prescribing information (? = not fully known).

**Table 1 pharmaceutics-17-00319-t001:** Cannabis-based products.

Cannabis-Based Products (Cannabis Preparations)				
No license for specific medical indications To use off-license for medicinal purposes Recommended quality control practices Approach to regulated cannabis-based products for medical use vary widely between countries				
Raw Cannabis	Compounding Preparations	Cannabis-Based Products with Unspecified Composition	Standardized Cannabis Based Medical Products *	
Unheated or “non-activated” cannabis	Patient-specific products based on a physician’s prescription prepared by a pharmacist	Flowers Tincture Chewable Lozenge Oil Infusions Cream Crystal Sublingual drop Balm Salve Lotion Spray Ointment	Inflorescences	Bedrocan^®^ (THC approx. 22%, CBD < 1%) Bedica^®^ (THC approx. 14%, CBD < 1%) Bedrobinol^®^ (THC approx. 13.5%, CBD < 1%) FM1 (THC 13–20%, CBD < 1%) FM2 (THC 5–8%, CBD7–12%) Bediol^®^ (THC 6.3%, CBD 8%) Bedrolite^®^ (THC < 1%, CBD 9%) Tilray THC 25^®^ (THC 25%, CBD < 1%) Tilray THC 10^®^ (THC 10%, CBD 10%)
			Cannabis extract diluted in oil	T10/C2^®^ (THC 10%, CBD 2%) T15/C3^®^ (THC 15%, CBD3%) T10/C10^®^ (THC 10%, CBD 10%) T5/C10^®^ (THC 5%, CBD 10%) T3/C15^®^ (THC 3%, CBD15%) T1/C20^®^ (THC1%, CBD 20%)

* Some examples (not all are listed).

**Table 2 pharmaceutics-17-00319-t002:** Cannabinoid prescription drugs.

Prescription Drugs					
Terminology	Cannabis-Derived Products		Cannabis-Related Products		
Origin	Compounds occurring naturally in the plant that are extracted directly from the plant		Synthetic compounds created in a laboratory and can be used to manufacture drug products		
Brand name	Epidiolex^®^	Sativex^®^	Marinol^®^	Syndros^®^	Cesamet^®^
Active principle/substance	CBD	Nabiximols (THC: CBD equal quantities)	Dronabinol	Dronabinol	Nabilone
Pharmaceutical form	Oral solution	Oromucosal spray solution	Oral capsules (Marinol^®^, Cesamet^®^) Oral solution (Syndros^®^)		
Indication	Patients 2 years of age or older with Dravet syndrome. Seizures associated with Lennox–Gastaut syndrome.	Muscle spasticity associated with multiple sclerosis.	Anorexia associated with weight loss in patients with acquired immunodeficiency syndrome. Vomiting and nausea associated with cancer chemotherapy in patients who have failed to respond to conventional treatments.		

**Table 3 pharmaceutics-17-00319-t003:** Cannabinoid medication (precipitant) affecting the metabolism of substrate medication (object) with narrow therapeutic index (NTI), based on the medication label.

Cannabinoid (Precipitant)	CBD (Epidiolex^®^)	Nabiximols (Sativex^®^, THC: CBD Equal Quantities)	Dronabinol (Marinol^®^ and Syndros^®^, Synthetic Form of THC)	Nabilone (Cesamet^®^, Synthetic Cannabinoid)
**Can affect the metabolism of substrate medication (object) with a NTI**	Inhibits and induces: CYP1A2 and CYP2B6 substrates Inhibits: CYP2C9, CYP2C19, CYP2C8, UGT1A9, and UGT2B7 substrates	Inhibits: CYP3A4, UGT1A9, UGT2B7 substrates Induces: CYP2B6, CYP1A2, and CYP3A4 substrates	CYP2C9 and CYP3A4 substrates	Weak inhibitor: CYP3A4 and CYP2E1 substrates Moderate inhibitor: CYP2C8 and CYP2C9 substrates

**Table 4 pharmaceutics-17-00319-t004:** Medications with a narrow therapeutic index (NTI) and cannabinoid medications with high protein binding. Medications in bold have both an NTI and high protein binding. Adapted from Kocis and Vrana (2020).

Medication with Narrow Therapeutic Index (NTI)			
Acenocuomarol	Dihydroergotamine		Mephenytoin
Alfentanil	Diphenadione		Mycophenolic acid
Aminophylline	Dofetilide		Nortriptyline
Amiodarone	Dosulepin		Paclitaxel
Amitriptyline	Doxepin		Phenobarbital
**Amphotericin B**	Ergotamine		Phenprocoumon
Argatroban	Esketamine		**Phenytoin**
Busulfan	Ethinyl estradiol (oral contraceptives)		Pimozide
Carbamazepine	Ethosuximide		Propofol
Clindamycin	Ethyl biscoumacetate		**Quinidine**
Clomipramine	Everolimus		Sirolimus
Clonidine	Fentanyl		**Tacrolimus**
Clorindione	Fluindione		Temsirolimus
Cyclobenzaprine	Fosphenytoin		Theophylline
**Cyclosporine**	Imipramine		Thiopental
Dabigatran etexilate	**Levothyroxine**		Tianeptine
Desipramine	Lofepramine		Trimipramine
Dicoumarol	Melitracen		**Valproic acid**
Digitoxin	Meperidine		**Warfarin**
**Cannabinoid Medication with Protein Binding ≥ 85% ***			
Cannabidiol		Nabilone	
Dronabinol		Nabiximols	

* Not considered an NTI.

## Data Availability

Not applicable.

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
