# Peer review of "An Overview of the Potential for Pharmacokinetic Interactions Between Drugs and Cannabis Products in Humans"

_pharmaceutics, 2025, doi:10.3390/pharmaceutics17030319_

Round 1
Reviewer 1 Report
Comments and Suggestions for Authors
In this comprehensive review, the authors discuss the drug metabolism of cannabinoids and the resulting pharmacokinetic interactions with other prescription drugs and drugs of abuse.
I have the following comments and suggestions:
Line 246: “Following THC consumption, urine predominantly contains acidic metabolites, such as THC-COOH…” How about the glucuronides?
Line 388: Table 3. “Inhibitis” should be “inhibits”
Line 401: The authors discuss the metabolism of dronabinol separately from that of THC. However, dronabinol is THC, albeit synthetic versus purified from pant material. The same molecule will be metabolized the same way. If the authors have any evidence that dronabinol indeed is potentially metabolized differently, this needs to be discussed.
Line 465: “CBD may increase stiripentol levels of 28% for Cmax…” should read “…levels by 28%...”
Line 505: 3.1.6. Transplant medications. Since these drugs are also used for the treatment of autoimmune diseases and, in the case of mTOR inhibitors, as proliferation inhibitors in cancer and cardiovascular drug-eluting devices, the header should read “Immunosuppressants and Proliferation Inhibitors”.
Line 573: “CBD may increase may increase caffeine exposure…” Please delete the duplicate “may increase”.
Table 4: Methadone: “CBD can inhibits methadone metabolism…” Should be: “CBD can inhibit..”
Table 4: Tacrolimus: “Adjust tacrolimus dosage and monitor plasma levels of tacrolimus.” Tacrolimus is monitored in whole blood and not in plasma.
Table 4: Sirolimus: “Adjust sirolimus dosage and, if it is possible, monitor plasma levels.” Sirolimus is monitored in whole blood and not in plasma.
Table 4: Everolimus: “Adjust everolimus dosage and monitor plasma levels.” Everolimus is monitored in whole blood and not in plasma.
Table 4: Cyclosporine: “Adjust cyclosporine dosage and if it is possible monitor plasma levels” Cyclosporine is monitored in whole blood and not in plasma.
Table 4: Metronidazole: “Dronabinol contains alcohol…” Dronabinol is the name of synthetic THC and does not contain alcohol. What the authors refer to here is the Dronabinol formulation Marinol. Please clarify. Overall, there are many statements regarding Dronabinol throughout the manuscript that are confusing. If referring to the API, please consistently refer to Dronabinol/THC, if to the formulated drug, please refer to Marinol.
Author Response
Reviewer 1
Comments and Suggestions for Authors
In this comprehensive review, the authors discuss the drug metabolism of cannabinoids and the resulting pharmacokinetic interactions with other prescription drugs and drugs of abuse. I have the following comments and suggestions:
- Line 246: “Following THC consumption, urine predominantly contains acidic metabolites, such as THC-COOH…” How about the glucuronides?
Response: As recommended, we have added a mention of the predominant urinary elimination of THC in its glucuronidated forms (THC-COOH-glucuronide). See line 270-271: “Following THC consumption, urine predominantly contains acidic metabolites, such as THC-COOH, mainly in its glucuronidated form…”.
- Line 388: Table 3. “Inhibitis” should be “inhibits”
Response: We have corrected the typographical error in Table 3 (page 11).
- Line 401: The authors discuss the metabolism of dronabinol separately from that of THC. However, dronabinol is THC, albeit synthetic versus purified from pant material. The same molecule will be metabolized the same way. If the authors have any evidence that dronabinol indeed is potentially metabolized differently, this needs to be discussed.
Response: You are right. In section 1.4.3, we discuss the metabolism of natural THC and CBD, and in section 3, where we specifically address dronabinol, we have removed redundant information on THC metabolism that was already covered in section 1.4.3. As a result, section 3 now contains a shorter and more specific paragraph focused on interactions. (See line 459-465).
- Line 465: “CBD may increase stiripentol levels of 28% for Cmax…” should read “…levels by 28%...”
Response: We have corrected the wording error, replacing "of" with "by". See line 527: "CBD may increase stiripentol levels by 28% for Cmax and 55% for AUC..."
- Line 505: 3.1.6. Transplant medications. Since these drugs are also used for the treatment of autoimmune diseases and, in the case of mTOR inhibitors, as proliferation inhibitors in cancer and cardiovascular drug-eluting devices, the header should read “Immunosuppressants and Proliferation Inhibitors”.
Response: As recommended, we have replaced "transplant medications" with "Immunosuppressants and Proliferation Inhibitors" in section 3.1.6 (line 569) and in Table 5 (page 19).
- Line 573: “CBD may increase may increase caffeine exposure…” Please delete the duplicate “may increase”.
Response: We have removed the redundancy in the wording. See line 642: "CBD may increase caffeine exposure by 15% for Cmax."
- Table 4: Methadone: “CBD can inhibits methadone metabolism…” Should be: “CBD can inhibit..”
Response: We have made the necessary correction, removing the extra "s". See Table 5, page 18: "CBD can inhibit methadone metabolism."
- Table 4: Tacrolimus: “Adjust tacrolimus dosage and monitor plasma levels of tacrolimus.” Tacrolimus is monitored in whole blood and not in plasma.
Response: We have corrected the error in the recommendation. See Table 5, page 19: "Adjust tacrolimus dosage and monitor blood levels of tacrolimus."
- Table 4: Sirolimus: “Adjust sirolimus dosage and, if it is possible, monitor plasma levels.” Sirolimus is monitored in whole blood and not in plasma.
Response: We have corrected the error in the recommendation. See Table 5, page 19: "Adjust sirolimus dosage and, if it is possible, monitor blood levels."
- Table 4: Everolimus: “Adjust everolimus dosage and monitor plasma levels.” Everolimus is monitored in whole blood and not in plasma.
Response: We have corrected the error in the recommendation. See Table 5, page 19: "Adjust everolimus dosage and monitor blood levels."
- Table 4: Cyclosporine: “Adjust cyclosporine dosage and if it is possible monitor plasma levels” Cyclosporine is monitored in whole blood and not in plasma.
Response: We have corrected the error in the recommendation. See Table 5, page 19: "Adjust cyclosporine dosage and, if possible, monitor blood levels."
- Table 4: Metronidazole: “Dronabinol contains alcohol…” Dronabinol is the name of synthetic THC and does not contain alcohol. What the authors refer to here is the Dronabinol formulation Marinol. Please clarify. Overall, there are many statements regarding Dronabinol throughout the manuscript that are confusing. If referring to the API, please consistently refer to Dronabinol/THC, if to the formulated drug, please refer to Marinol.
Response: As recommended, we have clarified when referring to dronabinol in general (the synthetic form of THC) (see line 225 and 459) and when referring to a specific formulation, such as the liquid formulation of dronabinol (Syndros®), which contains alcohol (see line 463 and Table 5, pages 21 and 23). We have also specifically clarified the pharmaceutical form of the synthetic formulations of cannabinoid prescription drugs in table 2 (See page 3)
Reviewer 2 Report
Comments and Suggestions for Authors
An overview of the potential for pharmacokinetic interactions between drugs and cannabis products in humans
This article is quite intriguing due to the expansion of the use of a previously prohibited plant, which has now broadened its application for both recreational consumption and therapeutic purposes. The review is concise and well-written and highlights the latest contributions from the literature concerning drug interactions with the main compounds of cannabis.
Some suggestions:
Line 33: Scientific names should be italicized (please check throughout the document).
In the section on ADME properties, the authors are encouraged to include some images or illustrations to enhance the text's dynamism for readers.
Line 341: The paragraph displays a different font size.
Author Response
Reviewer 2
Comments and Suggestions for Authors
An overview of the potential for pharmacokinetic interactions between drugs and cannabis products in humans
This article is quite intriguing due to the expansion of the use of a previously prohibited plant, which has now broadened its application for both recreational consumption and therapeutic purposes. The review is concise and well-written and highlights the latest contributions from the literature concerning drug interactions with the main compounds of cannabis.
Some suggestions:
- Line 33: Scientific names should be italicized (please check throughout the document).
Response: We have reviewed the entire manuscript and made the necessary corrections.
Line 33: "Cannabis sativa L., an herbaceous…"
Line 35-36: "such as Cannabis sativa ssp. sativa, Cannabis sativa ssp. indica, Cannabis sativa ssp. ruderalis, and Cannabis sativa ssp. Afghanica."
Additionally, when referring to in vitro studies, we have changed the formatting to italic. See lines 420, 426, 451, 454, 458, and 515.
- In the section on ADME properties, the authors are encouraged to include some images or illustrations to enhance the text's dynamism for readers.
Response: As recommended, we have added two figures in the ADME section, specifically in section 1.4.3 on metabolism, illustrating the metabolism of THC and CBD in a simplified manner (see Figure 1 and Figure 2).
- Line 341: The paragraph displays a different font size.
Response: We have standardized the font size in the paragraph beginning at line 369.
Reviewer 3 Report
Comments and Suggestions for Authors
The manuscript aims to presents the review of pharmacokinetic interactions between the available CBD and THC formulations and other drug products. The manuscript presents the data acquired from literature review with respect to interactions with various drugs. The authors provided information on the mechanism of interactions and a recommendation on the therapeutic solutions. They also classified every interaction using UpToDate classification. The main strength of the manuscript are as follows: clearly presented and reviewed articles, well structured table presenting the data, a thorough introduction and high number of reported cannabis-drug interactions. The article is well written, the authors clearly explained the pharmacokinetic basis of various interactions and also included the risk of drug-drug of abuse interactions. I would raise some points that could be better addressed by authors:
1) The Introduction section does not clearly state the research gap the author aim to fill. In fact numerous reviews, including two really good papers on the topic have already been published in recent years: doi: 10.3389/fphar.2024.1282831 and https://doi.org/10.3390/jcm11051154. They are both systematic review with clearly presented and executed methodology. They classify cannabis interactions based on their relevance and probability and they clearly rate the quality of evidence that the source articles represent, while the current manuscript mixes both strong and low evidence articles. In the current manuscript the recommendations on solving cannabis-drug interactions are sometimes drawn based on weak evidence of the occurrence of interactions. The article of Nachnani also compares the dosage regimens reported in various studies and briefly describes the studied patient population making it more useful to the readers.
2) 63 out of 118 cited works are the articles older than 5 years. This is significant number, since many citations refer to databases or published documents, so overall there aren’t many new scientific articles cited.
3) The manuscript presents some data on drug-cannabis interactions that are well documented, explained and based on thorough investigations (e.g. section 3.1.9) that is intertwined with some information that is based on low-level of evidence studies (e.g. section 3.1.8 and 3.1.11) or drugs that are currently insignificant due to its low use in treatment of diseases (section 3.1.3). These findings should be categorized based on their relevance.
4) I think that it could be worth giving some examples of narrow therapeutic index drugs that exhibit interactions with cannabis in lines 378-384 or to highlight them in table 4.
5) There are some parts that are out of scope of this manuscript and should be omitted, e.g. the fragment on rabbits that were given ophthalmic THC into their rectums (lines 209-212) or the one on tax revenues from cannabis in the US (lines 121-128)
6) I think that the manuscript could use some figures, e.g. one showing the metabolism of CBD and THC (text in Section 1.4.3) and another showing the influence of CBD and THC on the activity of various enzymes (text in lines 413-435
I would also highlight some minor points:
1) In reference 44 year and volume numbers should be separated by a space
2) Line 604 contains a typo (‘unknow’).
3) Reference 94 lacks a date when document has been accessed
4) In line 52 the sentence ‘CB1R mainly found in the central nervous system’ should be changed to: ‘CB1R is mainly found in the central nervous system’
Author Response
Reviewer 3
Comments and Suggestions for Authors
The manuscript aims to presents the review of pharmacokinetic interactions between the available CBD and THC formulations and other drug products. The manuscript presents the data acquired from literature review with respect to interactions with various drugs. The authors provided information on the mechanism of interactions and a recommendation on the therapeutic solutions. They also classified every interaction using UpToDate classification. The main strength of the manuscript are as follows: clearly presented and reviewed articles, well structured table presenting the data, a thorough introduction and high number of reported cannabis-drug interactions. The article is well written, the authors clearly explained the pharmacokinetic basis of various interactions and also included the risk of drug-drug of abuse interactions. I would raise some points that could be better addressed by authors:
- The Introduction section does not clearly state the research gap the author aim to fill. In fact numerous reviews, including two really good papers on the topic have already been published in recent years: doi: 10.3389/fphar.2024.1282831 and https://doi.org/10.3390/jcm11051154. They are both systematic review with clearly presented and executed methodology. They classify cannabis interactions based on their relevance and probability and they clearly rate the quality of evidence that the source articles represent, while the current manuscript mixes both strong and low evidence articles. In the current manuscript the recommendations on solving cannabis-drug interactions are sometimes drawn based on weak evidence of the occurrence of interactions. The article of Nachnani also compares the dosage regimens reported in various studies and briefly describes the studied patient population making it more useful to the readers.
Response: These two articles are referenced throughout our manuscript (see references 53 and 65), and we agree that they are high-quality systematic reviews that clearly evaluate the quality of evidence. However, they omit aspects from the prescribing information (PI) which, although some interactions may seem irrelevant, we believe should not be overlooked. We consider that our article provides a more comprehensive review of different drugs, even though it includes some information with a lower level of evidence. As recommended, we have included a classification with the quality of evidence (strong, moderate, weak), and clinical relevance (major, significant, minor, lack of relevance), See section 2 (line 379-399) and table 5. Pharmacokinetic interactions described in the PI have been included, and we have added a "Risk Rating" column in Table 5, indicating whether the interaction is documented in the PI or not. It is important to note that this manuscript is not a systematic review.
- 63 out of 118 cited works are the articles older than 5 years. This is significant number, since many citations refer to databases or published documents, so overall there aren’t many new scientific articles cited.
Response: We have added six new references from the current year (2025), see references 27, 43, 57, 71, 83, and 93—and deleted some references (references 21 and 22 from the previous manuscript)
- The manuscript presents some data on drug-cannabis interactions that are well documented, explained and based on thorough investigations (e.g. section 3.1.9) that is intertwined with some information that is based on low-level of evidence studies (e.g. section 3.1.8 and 3.1.11) or drugs that are currently insignificant due to its low use in treatment of diseases (section 3.1.3). These findings should be categorized based on their relevance.
Response: Although some information comes from studies with a low level of evidence or may seem less relevant, we believe it should not be omitted since these effects are documented in the prescribing information of the product. For example, this applies to antipyrine (referenced in the Cesamet® and Marinol® PI) in section 3.1.3, tricyclic antidepressants (referenced in the Marinol®, Syndros®, and Cesamet® PI) in section 3.1.8, and proton pump inhibitors such as omeprazole (referenced in the Sativex® prescribing information) in section 3.1.11. Since prescription and therapeutic decisions are primarily based on PI, we do not consider omitting these sections. For better clarity, in the table 5, we have included a classification with the quality of evidence, and clinical relevance.
- I think that it could be worth giving some examples of narrow therapeutic index drugs that exhibit interactions with cannabis in lines 378-384 or to highlight them in table 4.
Response: As recommended, we have added a table listing narrow therapeutic index (NTI) medications and cannabinoids with high protein binding (see Table 4). Additionally, we have included complementary text in line 430: "Additionally, cannabinoids are highly protein-bound, which may lead to displacement interactions with other protein-bound drugs, potentially increasing the free fraction of NTI medications and altering their pharmacokinetics…"
And in line 435: "In Table 4, we present medications with NTI and cannabinoid medications with high protein binding."
- There are some parts that are out of scope of this manuscript and should be omitted, e.g. the fragment on rabbits that were given ophthalmic THC into their rectums (lines 209-212) or the one on tax revenues from cannabis in the US (lines 121-128)
Response: As recommended, we have removed the last paragraph of section 1.4.1 (previously lines 209-212), which discussed rectal bioavailability in rabbits for ophthalmic use. Additionally, we have removed the text regarding cannabis tax revenues in the U.S. from section 1.3, along with its respective references (previously references 21 and 22).
- I think that the manuscript could use some figures, e.g. one showing the metabolism of CBD and THC (text in Section 1.4.3) and another showing the influence of CBD and THC on the activity of various enzymes (text in lines 413-435)
Response: As recommended, we have added two graphics illustrating the main metabolism of THC and CBD (Figure 1 and Figure 2). Additionally, we have made modifications to the text in section 1.4.3 (lines 235-242) to provide greater clarity in interpreting these figures. Furthermore, we have included a graph showing the influence of CBD/Epidiolex, THC/dronabinol, nabiximols, and nabilone on the activity of different enzymes, based on prescribing information (Figure 3, page 12).
I would also highlight some minor points:
- In reference 44 year and volume numbers should be separated by a space
Response: We have corrected the spacing between the year and the volume. See reference 55 (previously reference 44)
- Line 604 contains a typo (‘unknow’).
Response: We have corrected the typographical error, replacing "unknow" with "unknown". See line 672.
- Reference 94 lacks a date when document has been accessed
Response: We have added the access date to reference 100 (previously reference 94).
- In line 52 the sentence ‘CB1R mainly found in the central nervous system’ should be changed to: ‘CB1R is mainly found in the central nervous system’
Response: We have modified the phrase by adding "is", resulting in: "CB1R is mainly found in the central nervous system…" See line 51.